# A Comprehensive Study of Lupin Seed Oils and the Roasting Effect on Their Chemical and Biological Activity

**DOI:** 10.3390/plants11172301

**Published:** 2022-09-02

**Authors:** Eman F. Al-Amrousi, Ahmed N. Badr, Adel G. Abdel-Razek, Karolina Gromadzka, Kinga Drzewiecka, Minar M. M. Hassanein

**Affiliations:** 1Department of Fats and Oils, National Research Center, Dokki, Cairo 12622, Egypt; 2Department of Food Toxicology and Contaminants, National Research Centre, Dokki, Cairo 12622, Egypt; 3Department of Chemistry, Poznan University of Life Science, ul. Wojska Polskiego 75, 60-625 Poznan, Poland

**Keywords:** sweet and bitter lupin oils, roasting lupin seeds, antioxidant, antimicrobial and antifungal activity, toxigenic fungi, nontraditional oil

## Abstract

The present investigation aimed to study the impact of roasting on the chemical composition and biological activities of sweet and bitter lupin seed oils. Lupin oils were extracted using petroleum ether (40–60) with ultrasonic assisted method. Lupin Fatty acids, phytosterols, carotenoids, and total phenolic contents were determined. In addition, antioxidant, antimicrobial, and antifungal activities were evaluated. The results showed a ratio between 7.50% to 9.28% of oil content in lupin seed. Unroasted (bitter and sweet) lupin oil contained a high level of oleic acid ω9 (42.65 and 50.87%), followed by linoleic acid ω6 (37.3 and 34.48%) and linolenic acid ω3 (3.35 and 6.58%), respectively. Concerning phytosterols, unroasted (bitter and sweet lupin) seed oil reflected high values (442.59 and 406.18 mg/100 g oil, respectively). Bitter lupin oil contains a high amount of phenolics, although a lower antioxidant potency compared to sweet lupin oil. This phenomenon could be connected with the synergistic effect between phenolics and carotenoids higher in sweet lupin oil. The results reflected a more efficiently bitter lupin oil against anti-toxigenic fungi than sweet lupin oil. The roasting process recorded enhances the antimicrobial activity of bitter and sweet lupin seed oil, which is linked to the increment in bioactive components during the roasting process. These results concluded that lupin oil deems a novel functional ingredient and a valuable dietary fat source. Moreover, lupin oil seemed to have antifungal properties, which recommended its utilization as a carrier for active-antifungal compounds in food products.

## 1. Introduction

Recently, it has been interesting to search for new sources of natural bioactive components. Lupins are particularly interesting because their oils have significant antioxidant properties and health benefits for humans and animals [1]. Lupins are members of the family *Leguminosae*. There are two forms of lupin, bitter and sweet varieties (differ in alkaloid). White lupin seeds are consumed as a snack, and bitter ones are subjected to prolonged soaking and washing to remove bitter alkaloids. Lupin seeds are a good source of essential micronutrients (Fe, Cu, Mn, Zn, B, Mo, Ni, and Cl). Macronutrients (N, P, K, Ca, Mg, and S, are essential for a wide variety of physiological and metabolic processes in the human body [2]. Bitter and sweet lupin seeds and their oils contain natural antioxidants, preventing diseases and promoting health [3].

The impact of these oils was linked to minor components of oil extracted from the seed. In previous work, as nontraditional oil, lupin seed oil (LSO) is characterized by higher amounts of unsaturated fatty acids group, oleic, and linoleic acids [4,5]. The LSO contains high amounts of minor bioactive components such as phytosterols (1200 mg/100 g). Concerning tocopherols, LSO had more than 90% γ-tocopherol [4]. Nontraditional oils positively inhibit the toxigenic-fungi contamination of food-pathogen fungi due to bioactive components in oil such as tocopherol, tocotrienol, and polyphenols [2,4]. Tocopherol, tocotrienol, antioxidants, and phenolic compounds are included in fungal inhibition [6,7]. Moreover, oil phytochemical and minor components provide antioxidant potency for oil application and provide anti-mycotoxicological properties for food production safety [8].

Roasting is a significant step in processing seeds to enhance their aroma, flavor, and color and improve their shelf life, digestibility, nutritional value, and antioxidant properties [9,10]. Zou et al. [11] roasted wheat germ at 180 °C for different times and found that the TPC and Millard reaction products increased, which improved oxidative stability. Roasting has other effects, such as breaking cell membranes, which help the movement of oil through the permeable cell walls, resulting in higher extraction oil yield [12]. This effect may be due to chemical reactions in the seeds that, in turn, can significantly modify the chemical composition of the oil. Furthermore, the roasting process can improve microbiological food safety troy toxins and pollutants in seeds [7].

Toxigenic fungi are a natural hazard source that may contaminate seeds during handling, storage, and marketing [13]. This hazard could have happened while microbial infection occurred by bacteria, fungi, or their metabolites. Contamination of food-plant materials could render them to be un-useful food. Plant extracts are known as sources of bioactives and minor components [14]; these extracts could be polar or non-polar, including oils and fats. Vegetable oils, including nontraditional types, were reported to have biological activities against toxigenic fungi [15].

The previous research studied the oil extraction from lupin seed without referring to biological activity changes due to the roasting process. Thermal activity for lupin seeds before oil extraction may increase the phytochemical and bioactive contents. There are a few studies on the bioactive components of lupin seed (roasted and unroasted) and antimicrobial, antifungal, and antioxidant activity. This research aimed to study the chemical composition and biological activity of sweet and bitter lupin seed oils as well as the effect of the roasting process on them. The antimicrobial and antifungal activities of these oils were also determined. The results of this investigation may discover new lupin seed applications.

## 2. Results

### 2.1. Proximate Analysis

The oil content, acid value (AV), peroxide value (PV), p-Anisidine value (p-AV), and total oxidation value (TOTOX) were represented in Table 1. The oil content in the sweet lupin oil (SLO), roasted and unroasted, is higher than that in bitter lupin oil (BLO). The roasting of both kinds significantly increased the oil content. This result may be due to the impact of the roasting process that destroys the cell barriers, causing many permanent pores, facilitating oil moving through the cell walls, and variation of the moisture content in the seeds. It was noticed from the results that the AV of oils increased during the roasting process; this may be due to the hydrolysis of triglycerides at a high temperature and the accumulation of free fatty acids. After the roasting process, the changes in the AV of extracted oil for the sweet or bitter lupin represented significant differences. The changes recorded in the PV and p-AV were also reported as increasing, and these changes were manifested in significant differences. The changes also can be illustrated due to the increase in primary and secondary oxidation products in the oil and, consequently, the increase in total oxidation value.

The calculated value of the TOTOX index for the investigated oils varied between 11.95 ± 1.49 to 35.3 ± 3.08, where the high value was recorded for the oil extracted from the roasted bitter lupin (RBLO). The borderline level determining good oil quality is TOTOX. Among the analyzed oils, all of them exceeded this limit. The RBLO achieved a great value for the TOTOX index; this point can be explained by the changes recorded in the values of the PV and p-AV. The lowest fat oxidation index was found for unroasted oil from the sweet lupin; however, the recorded number still exceeds the TOTOX standard limit.

### 2.2. Major Components of Fatty Acid Composition

The fatty acid composition of different LSO samples is represented in Table 2. Considering saturated fatty acid content (SFA), it was found that oil from unroasted bitter seeds has a higher content of SFA than oil from unroasted sweet seeds. The most abundant SFA are palmitic and stearic acids, which are higher in the unroasted oil of bitter seeds (8.89 and 3.52%, respectively). The primary monounsaturated fatty acid (MUFA) was oleic acid. It was higher in the oil from unroasted sweet seeds (50.87%) than that in the oil from unroasted bitter seeds(42.65%). Linoleic acid (ω6) as polyunsaturated fatty acid (PUFA) was found to be higher in the oil from unroasted bitter seeds than that in the oil from unroasted sweet seeds (37.38 and 34.48, respectively). In contrast, linolenic acid (ω3) was found to be higher in the oil from unroasted sweet seeds(6.58%) than in the oil from unroasted bitter seeds (3.35%) (Table 2).

The two types of seeds were roasted, and the extracted oil was classified as roasted bitter lupin oil (RBLO) and roasted sweet lupin oil (RSLO). The results in Table 2 show that the ratio of ω6/ ω3 ranged from 5:1 to 11:1. From the nutritional point of view, lupin seed oil has a desirable balance of ω-6 to ω-3 fatty acids. Erucic acid was not detected in the oil from unroasted bitter seeds but was found in a minimal amount in the oil from unroasted sweet seeds (0.25%). The total UFA was higher in the oil from unroasted sweet seeds (92.44%), while the total SFA was higher in the oil from unroasted bitter seeds (14.04%). The ratio of SFA/UFA of the oil from unroasted sweet and bitter lupin seeds were 0.081 and 0.154 respectively, showed that lupin oil is closer to corn, sesame, and quinoa oils. It can be concluded that the LSO is a good source of UFA (85.96 to 92.44 %). Oleic acid has a role in preventing some diseases such as atherosclerosis, thrombosis, and cancer. It is also a rich source of essential fatty acids (linoleic ω6 and linolenic ω3 up to 40%).

Roasting lupin seeds at 180°C for 10 min decreased the amount of most SFA, such as palmitic acid (in both the oil extracted from sweet and bitter seeds of lupin) and stearic acid (in unroasted bitter oil). In the oil from roasted sweet lupin, the ratio of stearic acid was slightly increased by roasting (Table 2). Oleic acid, the main MUFA, decreased from 50.87% to 48.43% for oil of roasted sweet lupin, while it increased from 42.65% to 46.53% for oil from roasted bitter lupin during the roasting process. These changes are often associated with the release of fatty acids from the oil due to heat treatment. This action may change the proportions of fatty acids and their distribution within the oil after roasting, as shown in Table 2.

Considering the PUFA, the value of ω6 was increased in RSLO, while ω3 was slightly decreased by roasting treatment. Subsequently, the ω6: ω3 ratio increased from 5:1 up to 6:1; while it still was in the favorable area (around 10:1). In contrast, the ω6 decreased, and ω3 increased by roasting the BLO. The ratio between ω6 and ω3 was changed from 11.16 to 6.8 after the roasting process (BLO). In general, the total SFA and total PUFA amount decreased by roasting in SLO and BLO, while the amount of essential fatty acids (ω6 + ω3) increased (Table 2). The results reflected a more saturated fatty acid content for bitter lupin oil (roasted and unroasted) than sweet lupin oil, close to the multiplied content in bitter oil. In contrast, the oil content of monounsaturated fatty acids recorded more values for sweet lupin oils. The same situation has been shown for total unsaturated fatty acids. In contrast, total polyunsaturated fatty acids values were quite close to oil extracted from roasted lupin of sweet and bitter seeds. In the oil evaluation process, some parameters are calculated as represented in Table 2; these parameters can indicate the significant changes that occurred to the oil, which may affect its quality and functionality properties. Otherwise, some parameters such as PUFA/SFA or L/Ln can indicate the better biological activities of oil by in vivo application.

### 2.3. Minor Bioactive Components

#### 2.3.1. Phytosterols Composition

Different members of the phytosterols group are represented in Table 3; the β-sitosterol was the most abundant phytosterol, followed by campesterol. The higher content of β-sitosterols was in USLO than in UBLO (326.57 and 303.8 mg/100 g oil, respectively). Concerning campesterol, it was found in a higher amount in UBLO (77.35 mg/100 g oil) than in other oil samples. The UBLO has the highest content of Δ 5 avenasterol (59.13 and 45.95 mg/100 g oil for the UBLO and the RBLO, respectively).

The total phytosterol content in UBLO was higher than that of USLO (442.59 and 406.18 mg/100 g oil, respectively). According to Table 3 results, a slight decrease occurred in total phytosterols and phytosterol composition due to the roasting process, except for campesterol in the oil from roasted seeds of sweet lupin, which was not affected.

#### 2.3.2. Carotenoid Composition

The data of carotenoids represented in Table 3 indicated that total carotenoid content in USLO (101.19 mg/100 g oil) was higher than that in the oil of unroasted bitter lupin (66.88 mg/100 g oil). β-carotene, pro-vitamin A, is the primary carotenoid in oil extracted from sweet lupin seeds; and it is higher in the oil from unroasted sweet lupin seeds than in the oil from unroasted bitter seeds (99.3 and 66.57 mg/100 g, respectively). Other carotenoids, such as lutein and zeaxanthin (an isomer of lutein), were higher in USLO than in the UBLO. The results in Table 3 showed the degradation of carotenoids during the roasting process. Both total carotenoid and β-carotene content were slightly decreased by roasting. Concerning β-carotene, it is little reduced by roasting from 99.3 to 97.97 mg/kg oil for RSLO and from 66.57 to 65.57 mg/kg oil for RBLO. In contrast, total carotenoid was decreased from 101.19 to 98.91 mg/kg oil for the RSLO and from 66.91 to 65.64 mg/kg oil for the RBLO.

### 2.4. Antioxidant and Biological Activity of Lupin Seed Oil

The antioxidant activity of oils depends on the type, abundance, and category of bioactive compounds. The antioxidant activity of different lupin oil samples and the effect of roasting on them were evaluated using the following two evaluation methods (DPPH• scavenging assay and β-carotene-linoleic acid oxidation method).

#### 2.4.1. DPPH• Scavenging Assay

The antioxidant activity of the oil sample was evaluated by its ability to scavenge the DPPH• free radicals. Figure 1 shows the radical scavenging activity (RSA %) of different amounts of oils (about 10, 20, 30, and 40 mg oil). From Figure 1, it was noticeable that the roasting process gave higher activity for USLO. The higher potency was provided by the RSLO followed by the RBLO, the USLO, and the UBLO, which indicates that the activity increased by roasting, which may be due to the increase in the TPC. We also calculated both EC50, the amount of oil that can scavenge 50% of DPPH•, and the antioxidant power of 1/EC50 (Table 3). As the EC50 decreased, the antioxidant capacity and activity increased. RSLO gave the lowest EC50 and higher activity. The antioxidant activity can be arranged to descend as follows: the RSLO> the RBLO> the USLO> the UBLO, where the present result agrees with the previous one [16].

#### 2.4.2. The β-Carotene-Linoleic Acid Oxidation Method (Coupled Autoxidation)

The ability of oils to decrease the oxidation of β-carotene and the color change was determined as antioxidant activity (AOA %). The highest (AOA %) was given by RSLO, followed by USLO and RBLO (76.7, 74.22, and 72.89%, respectively), while the lowest AOA% was provided by UBLO (66.63%). Although UBLO has a higher TPC than USLO, it was found to have a lower (AOA %), and the same results were observed for RBLO and RSLO (Table 3).

The relationships between TPC, TCC, and the antioxidant activity of different oil samples were studied, as illustrated in Figure 2. Generally, the BLO (roasted and unroasted) that contains a higher amount of TPC has lower antioxidant activity than the SLO (roasted and unroasted) for both the DPPH• scavenging assay and β-carotene-linoleic acid oxidation method. This impact may be attributed to the synergistic effect between TPC and the carotenoids, which were found in higher amounts in the USLO and the RSLO, increasing the activity of the bioactive compounds and giving higher antioxidant activity.

#### 2.4.3. Total Phenolic Content

The total phenolic content (TPC) represented in Figure 2 indicates that the UBLO has a more excellent value of TPC than USLO (479.3 and 413.63 mg gallic acid/100 g oil, respectively). The amounts of the TPCs in the RBLO and the RSLO increased by roasting the seeds (to 570.95 and 515.5 mg gallic acid equivalent/100 g oil, respectively).

The oil treatment, such as roasting, commonly serves minor components and the released phenolic acids. These bioactives are known to aid the antioxidant activity of oil content. As shown in Figure 2, the change in total phenolic content between roasted and unroasted seed oil was manifested during the roasting process. These increments were recorded linked to the enhancement in the antioxidant activity and antiradical power of the resulting oil after roasting, which links to the raising of total phenolic contents for oil of bitter and sweet lupin seeds. These properties can also aid the extension of oil shelf life and its biological activities.

#### 2.4.4. Antibacterial and Antifungal Activity

The impact of LSOs as antibacterial was evaluated in the presence of a standard antibiotic (ofloxacin) as a control reference. The results showed that the RLSOs (bitter or sweet) have higher antimicrobial activity than the oil of the unroasted sweet lupin seeds. Generally, the obtained results show a low antibacterial potency of applied LSOs against Gram-negative strains of bacteria (Salmonella senftenberg ATCC 8400 and Escherichia coli ATCC 11228). In contrast, Gram-positive (Staphylococcus aureus ATCC 33591 and Bacillus subtilis DB 100) strains reflect a moderate antibacterial effect (Table 4).

Considering antifungal activity, the results of the inhibition zone diameter (IZ) of the fungal growth are represented in Table 4. The results indicated the higher efficiency of UBLO antifungal activity than that in USLO; however, they were still far from the standard fungicidal (nystatin) effect as a control reference. This influence may be due to the high content of unsaturated fatty acids recognized as the degradable content of mycotoxin secretion in some fungal biological systems. The obtained results manifested an immediate antifungal effect of lupin oil types for applied Aspergillispecies, with more impact on the Fusarium strain. Although the antifungal activity of oil is manifested by a limited impact on agar media growth, it still possesses the capability to protect the seeds and extend their shelf-life.

It was noticed that the roasting process could significantly ameliorate the antimicrobial activity of lupin oil against some bacterial and toxigenic fungal strains. The improvement of antimicrobial can be joined with the increment that occurred in minor components released by roasting; however, the reduction that recoded significantly for some strains of toxigenic fungi may be connected to the losses that occurred in some ingredients by the roasting process.

## 3. Discussion

Regarding the increment in the request for novel bioactive components from plant materials, lupin seed is considered a fantastic source of nontraditional oil rich in phytochemicals and active ingredients. Two types of lupin seeds are known to be utilized in the middle-east area (bitter and sweet lupin); however, oils sourced from these lupin types could have differed in their bioactive contents. Implementing these oils, either as new oil or as bioactive material applied in food products, may achieve several objectives for safe and nutritional food production. In contrast, seed treatments, including roasting, may change the oil characteristics that result from treated seeds. The roasting process of plant seeds before oil extraction could be enhanced their yield [12].

According to the present results, both types of roasted seed oil are enhanced for their oil content. This enhancement might be due to the roasting process damaging cell barriers, resulting in many permanent holes that allow oil to move through the cell walls and a variance in the moisture content of the seeds [12]. The findings revealed that the AV of oils rose throughout the roasting process, which might be attributed to the hydrolysis of triglycerides at high temperatures and the formation of free fatty acids. Oxidation also enhanced PV and p-AV due to increased primary and secondary oxidation products in the oil, increasing the total oxidation value [12]. The results reported by Suri et al. [17] were shown to concur with these findings.

The acid value is defined as the number of milligrams of potassium hydroxide necessary to neutralize the free fatty acids in one gram of fat [18]. This phrase is known as a neutralization number or acid number. It is a relative measure of rancidity since free fatty acids are usually produced during the degradation of triglycerides [19]. Typically, the acid value is converted to an FFA content by multiplying the acid value by a factor equal to the molecular weight of the fatty acid in question (generally oleic acid) divided by ten times the molecular weight of the potassium hydroxide, which indicates the quality of lubricity or rate of degradation when stored over time, with implications for stability and shelf life. Lipid oxidation is a substantial deteriorative process with considerable consequences for the quality and value of fats and oils, particularly the off-flavors caused by autoxidation. Hydroperoxides accumulate as main oxidation products during the early stages of the oxidation process, breaking down to generate low molecular weight oxygenated components such as alcohols, aldehydes, free fatty acids, and ketones, resulting in rancidity [19]. The presence of natural bioactive components assisted in protecting the oil against damage factors. The measurement of peroxide value is often used to monitor the buildup of hydroperoxides. The PV and the AV, a measure of hydrolytic rancidity, are two of the most often measured quality indicators throughout oil production, storage, and sale. Carbonyl molecules are more stable than hydroperoxide and may be used to assess oxidative changes in lipids [20].

According to Fontanari et al. [21], lupin oil has a lipid profile comparable to other sources of oil, such as olive oil, and has a high potential for human consumption. Their chromatographic analyses show that monounsaturated oleic fatty acid dominates in lupin seed oil, as demonstrated by the present study. The research of Fontanari [21] also mentioned the processing of lupin products, mainly extracted oil, to allow for use in food and other potential applications.

The variation in fatty acid content between the two types of lupin (bitter and sweet), besides their roasted seed oil, is reflected in the distinguished content of omega fatty acids. Ref. [4], reported that palmitic acid in lupin seed oil amounted to 7.5%. Kris-Etherton et al. [22] found the effect of SFA in decreasing cardiovascular disease risk reduction. Linoleic acid (ω6) was more significant in the UBLO than in the USLO as a polyunsaturated fatty acid (PUFA). At the same time, linolenic acid (ω3) was found to be greater in the USLO than in the UBLO (Table 2). Khalid and Elharadallou [16] reported close results in the same direction as the same conclusions. According to Khalid and Elharadallou [16] and Hassanein et al. [4], lupin seed oil has a significant concentration of w3, ranging from 9.95 %to 14.9%. The two necessary fatty acids (C18:2 and C18:3), which the human body cannot produce, must be supplied from food. These two fatty acids compete for the same enzyme, which transforms fatty acids into physiologically active forms [21]. The investigation of Hu [23] proposed that a linoleic to linolenic acid (ω6/ω3) ratio of 10:1 or below may prevent cardiovascular disease. Previous research has shown that the appropriate balance of ω6/ω3 varies depending on the ailment under consideration, and the dosage of 3 depends on the disease’s severity; however, a lower ratio of w6/w3 is preferred to lower chronic illness risk [24].

In a previous study, oligosaccharides’ content was reported significantly in lupin seed flour (of bitter and sweet lupin). There was a significant difference in both oils’ saponification number and ester value for the oils extracted from the sweet and bitter lupin seeds [16]. The peroxide value was found lesser in bitter oil than in sweet oil. Fatty acid composition showed higher values of total unsaturated fatty acids for bitter oil than sweet lupin oil; however, that study also referred to the differences between the two types of lupin oil. The steroids, triglycerides, and alcohols distinguished oil from sweet lupin seeds. At the same time, bitter seed oil was distinct by a high content of mono and di-glycerides, phospholipids, and hydrocarbons. In general, the essential fatty acids in USLO (41.06%) were somewhat more significant than in the UBLO (40.65%). These findings corresponded with Alamri et al. [25], who found 41.9% oleic acid in lupin seed oil. Erucic acid was not discovered in the UBLO but was identified as trace amounts in the USLO (0.25%).

The presence of ω6 can neutralize SFA, prevent cholesterol accumulation in the blood vessels, improve utilization of soluble vitamins, and govern nervous system functions [16]. In addition, from the nutritional point of view, USLO and UBLO have the desired ratio of ω6/ω3 fatty acids value. Karupaiah and Sundram (2013) found that decreases in PUFA/SFA ratio in human food were associated with an increased postprandial level of HDL-C in plasma. It was noticeable from data in Table 2 that USLO has a low proportion of PUFA/SFA, which is favored for human health [26].

A Cox value is a calculated value based on the percentage of UFA present in the oil, and it is usually taken as an indicator of the oil’s tendency to oxidation [27]. The Cox value is an indication of oil oxidation stability. The lower the cox values, the higher the oil oxidation stability [28]. The Cox values of different lupin oil samples were approximately equal, ranging from 4.96 to 5.50. The roasting process caused little increase in the Cox value, as shown in Table 2. These Cox values are observed to be better than some other traditional and nontraditional edible oils such as soybean oil (6.89), sunflower oil (6.42) (Hassanein et al., 2016), black cumin seed oil (6.6), grape seed oil (7.3), tomato seed oil (6.5), and wheat germ oil (7.8) [29]. These results indicate the high stability of lupin seed oils (roasted and unroasted) compared to some oils.

The UBLO contains the largest concentration of Δ 5 avenasterol (59.13 mg/100 g oil for the UBLO and 45.95 mg/100 g oil for the RBLO, respectively), which has a variety of therapeutic qualities including anticancer, antioxidant, anti-inflammatory, antidiabetic, neuroprotective, hepatoprotective, and so on [30]. This concentration of Δ 5 avenasterol was comparable to flaxseed oil (56.01 mg/100 g oil) and rapeseed oil (40.92 mg/100 g oil) [31]. The UBLO has a greater total phytosterol concentration than the USLO (442.59 and 406.18 mg/100 g oil, respectively). According to Table 3, roasting induced slight declines in total phytosterols and phytosterol composition, except for campesterol in the RSLO, which was unaffected by the roasting procedure [12].

Gutiérrez-Rosales et al. [32] postulated that carotenoids act as protectors that capture free radicals, similarly to α-tocopherol. The main actions of carotenoids are antioxidant and anti-inflammatory properties, which are essential for eye health [33]. They also effectively reduce insulin resistance and lipid accumulation [34].

Carotenoids are expected to degrade quickly by roasting, which may be due to their heat-sensitive properties [12].

The effect of the roasting process on TPC depends on the balance between the thermal degradation of bioactive compounds and the formation of new ones. The increase in the TPCs may be due to the higher ratio of newly formed polyphenols to thermally degraded phenols during the roasting process [12]. These phenolic compounds are common bioactive compounds that have antioxidants, are antimicrobial, anti-allergenic, and protect against many diseases [35]. Regarding Abdel-Razek et al. [36] investigation, they reported that carotenoids, together with polyphenols and tocopherols, support the oxidative stability of the oil. The high antioxidant contents of polyphenols, carotenoids, and tocopherols make lupin seed oil suitable for cosmetics, pharmaceuticals, and natural food additives in different food products [35].

Roasting can also foster the stability of oils against oxidation by deactivating enzymes such as lipases and lipoxygenases that can induce fatty acid oxidation and microbial destruction [37]. Although roasting can cause the degradation and oxidation of bioactive compounds such as phenolic compounds, some of the phenolic compounds can be incorporated into melanoidins, which have antioxidant proprieties related to their metal chelating capacity and their phenolic moieties [12,38]. The antioxidant activity can be arranged to descend as follows: the RSLO > the RBLO > the USLO > the UBLO, where the present result agrees with the previous one [16].

Specific fatty acids may behave as sporogenic agents for toxigenic fungi, enhancing spore production (vegetative development) and releasing less poison throughout the fungi’s life cycle. Otherwise, the impact of nontraditional oils containing bioactives could be extended to affect the metabolic lifecycle of toxigenic fungi. Tocols (tocopherol and tocotrienol) were another class that showed potential defense versus mycotoxin oxidative stress in live fungi systems, resulting in decreased mycotoxin production [39]. Otherwise, phytochemicals and phytosterols could provide safety properties through their applications in safe food production [40]. More investigations are required to evaluate the impact of lupin oils against mycotoxin production by toxigenic fungi, particularly on gene expression or molecular docking; however, higher antimicrobial and antifungal activities were found in the UBLO and the RBLO. This impact may be attributed to quinolizidine alkaloids that make it toxic for microorganisms such as bacteria and fungi, which are considered a natural source of hazard that may contaminate seeds during handling, storage, and marketing [41]. The present results could be recommended lupin oil application in food processing for several purposes. These purposes include nutritional applications, safety applications against harmful microbes, or the implementation as a carrier for food safety applications targeted to enhance safety properties and extend shelf life.

## 4. Materials and Methods

### 4.1. Materials

All solvents are analytical grade and were purchased from El-Nassr Pharmaceutical Chemicals Co. (ADWIC), Egypt. The Folin-Ciocalteu reagent was purchased from Sisco Research Laboratories Chemicals, India. The DPPH· from Sigma-Aldrich (St Louis, MO, USA). Bitter lupin (Lupinus termis) and sweet lupin (*Lupinus albus*) seeds were purchased from the local market in Giza city, Egypt.

Applied strains of pathogenic bacteria were purchased from the American type culture collection, LGC Standards, Queens Road, United Kingdom. These strains were *Salmonella senftenberg* ATCC 8400, *Escherichia coli* ATCC 11228, *Staphylococcus aureus* ATCC 33591, and *Bacillus subtilis* DB 100. The stock cultures were maintained by the nutrient agar (4 °C).

The strains of toxigenic fungi were purchased from the agro-food microbial culture collection, institute of sciences of food production, Italy. Strains of fungi were maintained by the malt extract agar media (2 °C). Fungi strains of the experiment were *Aspergillus flavus* ITEM 698, *A. parasiticus* ITEM 11, *A. carbonarius* ITEM 5010, *A. ochraceous* ITEM 282, and *Fusarium culmorum* KF 846.

### 4.2. Methods

#### 4.2.1. Preparation of Lupin Seed and Extraction of Oil

Lupin seeds (sweet and bitter) were cleaned and rendered free of dust and foreign bodies; each kind was divided into two parts. One part of each kind was roasted in the oven at 180 °C for 10 min. All seeds were then crushed to a fine powder using a mixer lab machine. The oil was extracted from seed samples twice by sonication for one hour using petroleum ether 40–60 (250 mL solvent/50 g seeds). The solvent was vacuum evaporated using a Heidolph rotary evaporator; the collected oils were kept in the refrigerator at 4 °C in the amber bottle till the analysis.

#### 4.2.2. Determination of Oil Content

The oil content was determined by extracting oil from a 100 g lupin seed sample using petroleum ether 40–60 in the soxhlet apparatus [16,42]. Acid value (AV) was determined using the AOCS official method Cd 3a–63, which was described by Posada et al. [43]. Peroxide value (PV) was determined according to AOCS’s official method Cd 8b-90 [44]. P-anisidin value (p-AV) was determined as described in detail by El-Mallah et al. [45]. The total oxidation Value (TOTOX value) is the sum of PV and p-AV used to estimate the total oxidative deterioration of the oil. It was calculated according to the equation:(1)TOTOX value = p−AV +2 PV 

#### 4.2.3. Determination of Fatty Acid Composition

Fatty acid methyl esters (FAME) were prepared according to AOCS Official Method Ce 1k-07 as described in our previous work (Abdel-Razek et al., 2021). The oil s oxidizability (COX) value was calculated by the formula proposed by Fatemi et al. [29].
(2)$$\mathrm{Cox}\;\mathrm{value}=\frac{1\times \left[C18:1\left(\%\right)\right]+10.3\times \left[C18:2\left(\%\right)\right]+21.6\times \left[C18:3\left(\%\right)\right]}{100}$$

#### 4.2.4. Determination of Bioactive Components

##### Determination of Phytosterol Composition

Regarding the methodology described by Stuper-Szablewska et al. [46], the phytosterol content indicated a minor component of the examined oil.

##### Determination of Carotenoids Composition

The carotenoid composition was evaluated utilizing the Acquity ultra-high performance liquid chromatography (Waters, Milford, MA, USA) as reported by Stuper-Szablewska et al. [47]. Carotenoids were determined as an indication of their activity as bioactive components.

##### Determination of Total Phenolic Content

Total phenolic content in oil was determined using Folin-Ciocalteu colorimetric method by UV-visible spectrophotometer using Shimadzu, UV-spectrophotometer UV-240 [46].

#### 4.2.5. Measurement of Antioxidant Activity

##### DPPH Radical Scavenging Activity

DPPH• assay was used to determine the antioxidant activity of oil samples using Toluene which can dissolve both oil samples and DPPH. The EC50 (concentration of sample that can scavenge 50% of DPPH• radicals) was determined by plotting the concentration against the RSA % on the excel program. The relation line was drawn, and the EC50 was calculated from the resulting equation described in our previous work [45].

##### β-Carotene-Linoleic Acid Oxidation Method (Coupled Autoxidation)

Antioxidant activity was also determined by the β-carotene-linoleic acid bleaching method (coupled autoxidation). The spectrophotometric method is based on the different extracts’ ability to decrease the oxidative losses of β-carotene in a β-carotene/linoleic acid emulsion [48].

#### 4.2.6. Determination of Antibacterial and Antifungal

The impact of lupin oil on pathogenic bacteria and fungi was determined according to Kaya et al. [49]. Quantities of 100 µL oil were injected into agar-well diffusion in the presence of tested bacterial or fungal strains. The inhibition zone diameter for each oil type was measured in millimeters against the control bacterial and fungal growth plate. The more the diameter zone, the higher the inhibition effect of the oil.

#### 4.2.7. Statistical Analysis

All investigations were performed in triplicate; the results were subjected to a one-way analysis of variance (ANOVA), followed by Tukey’s test using SPSS software (ver., 16, IBM, Armonk, NY, USA). The significance level was (*p* = 0.05).

## 5. Conclusions

The obtained results confirmed that the lupin seed oil was abundant in many bioactive components, oleic acid (ω-9) at about 50%, linoleic acid (ώ-6) at about 35%, and linolenic (ω-3) from 3–6%. The ω-6/ω-3 ratio ranged from 5–11, still in the excellent range (around 10:1). The oil extracted from unroasted bitter lupin seeds was rich in total phytosterols, total phenolic content, and consequently had more efficient antioxidant activity than the oil extracted from unroasted sweet lupin seeds. In addition, the oil of unroasted sweet lupin seeds contained high amounts of total carotenoids. It was found that oil from roasted bitter lupin seeds was rich in total phytosterols and total phenolic content, and had higher antioxidant activity than oil of roasted sweet lupin seeds. This result means that the roasting process at 180 °C for a short time (10 min) enhanced the minor bioactive components (phytosterols, TPC, TCC). In addition, the roasting process enhanced the antimicrobial activity of the oil extracted from the bitter and sweet lupin seeds. It was noticeable that the oil extracted from unroasted bitter lupin seeds was more efficient in antifungal activities than the oil extracted from unroasted sweet lupin seeds.

Consequently, the RLSO can be used to maintain food as safe from some microorganisms due to its increasing antimicrobial and antioxidant activity. One could say the lupin seed oil could be considered an excellent novel source for functional food ingredients and may be a valuable source of dietary fat and be value-added for different food and non-food purposes. In addition, it can be used to avoid or minimize the use of harmful synthetic additives.

## Figures and Tables

**Figure 1 plants-11-02301-f001:**
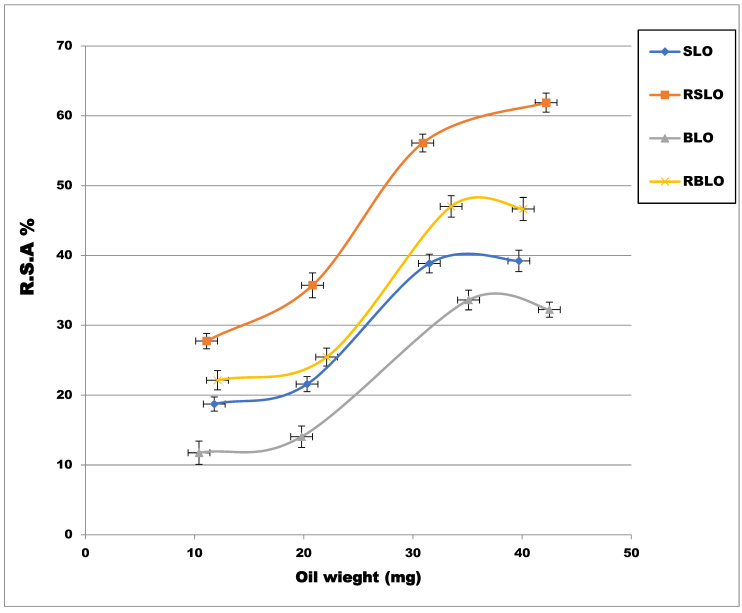
Radical scavenging activity (RSA %) of different lupin oil samples. The changes in values for each line, which represented different lupin oil types, were significant differences.

**Figure 2 plants-11-02301-f002:**
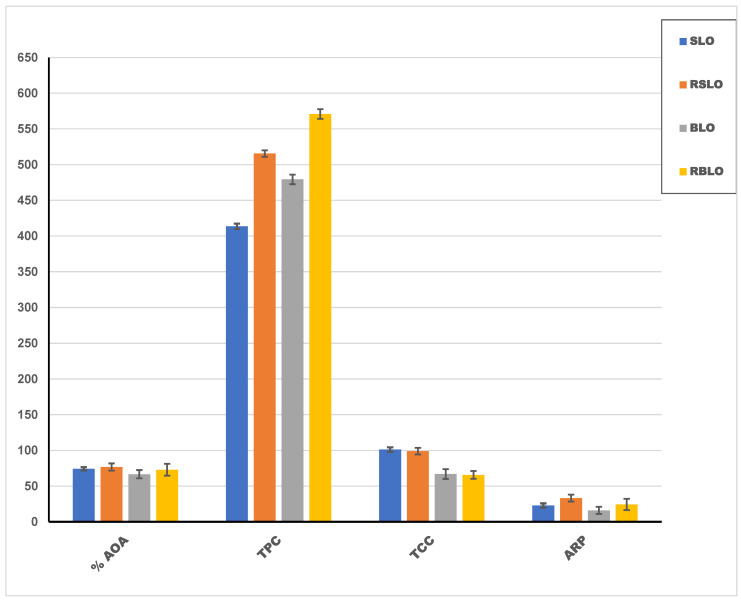
The relation between phenolic, carotenoid, antioxidant activity contents, and the antiradical power of different lupin oil samples. TPC: total phenolic content; TCC: total carotenoid content; %AOA:.antioxidant activity; ARP: antiradical power [(1/EC50) × 1000].

**Table 1 plants-11-02301-t001:** Proximate analysis of unroasted and roasted sweet and bitter lupin seed oils.

Proximate Analysis	Lupin Seeds Oil Characteristics			
	Sweet		Bitter	
	Unroasted	Roasted	Unroasted	Roasted
**Oil content %**	8.92 ± 0.21 ^a^	9.28 ± 0.34 ^b^	7.50 ± 0.05 ^c^	7.85 ± 0.11 ^d^
**AV**	0.75 ± 0.05 ^a^	1.06 ± 0.003 ^b^	1.1 ± 0.01 ^c^	1.41 ± 0.07 ^d^
**PV**	3.89 ± 0.94 ^a^	10.44 ± 1.5 ^b^	5.44 ± 1.19	10.85 ± 2.26
**p-AV**	4.17 ± 0.5 ^a^	5.99 ± 0.8 ^b^	4.82 ± 1.17 ^c^	13.6 ± 0.75 ^d^
**TOTOX value**	11.95 ± 1.49 ^a^	26.87 ± 1.133 ^b^	15.7 ± 2.37 ^c^	35.3 ± 3.08 ^d^

AV = acid value, PV = Peroxide value, p-AV = P anisidine value, TOTOX value = total oxidation value. The values with different letters in the same row had significant differences.

**Table 2 plants-11-02301-t002:** Fatty acid composition of unroasted and roasted sweet and bitter lupin seeds oil.

Oil Parameters	Sweet Lupin Seeds Oil		Bitter Lupin Seeds Oil	
	Unroasted	Roasted	Unroasted	Roasted
**Saturated fatty acids**				
C16:0 (palmitic acid)	6.26 ± 0.06 ^c^	5.61 ±0.37 ^d^	8.89 ± 0.21 ^a^	6.95 ± 0.35 ^b^
C17:0 (heptadecanoic acid)	ND	0.17 ± 0.29 ^c^	0.77 ± 0.13 ^a^	0.2 ± 0.26 ^b^
C18:0 (stearic acid)	0.61 ± 0.18 ^d^	1.77 ± 0.45 ^c^	3.52 ± 0.15 ^a^	2.53 ± 1.24 ^b^
C20:0 (arachidic acid)	ND	0.31 ± 0.45 ^c^	0.82 ± 0.12 ^a^	0.6 ± 0.34 ^b^
C21:0 (heneicosanoic acid)	0.05 ± 0.08 ^a^	ND	0.04 ± 0.05 ^b^	ND
C24:0 (tetracosanoic acid)	0.64 ± 0.21 ^a^	0.1 ± 0.05 ^b^	ND	ND
Total SFA	7.56 ± 0.53 ^c^	7.94 ± 1.16 ^c^	14.04 ± 0.66 ^a^	10.28 ± 2.19 ^b^
**monounsaturated fatty acid**				
C16:1 (palmitoleic acid)	0.26 ± 0.07 ^d^	0.7 ± 0.17 ^c^	1.93 ± 0.14 ^a^	0.99 ± 0.54 ^b^
C17:1 (heptadecenoic acid)	ND	ND	0.2 ± 0.04	ND
C18:1 (oleic acid) ω9	50.87 ± 0.37 ^a^	48.43 ± 1.14 ^b^	42.65 ± 0.29 ^d^	46.53 ± 1.05 ^c^
C20:1 (eicosenoic acid)	ND	0.1 ± 0.07 ^c^	0.49 ± 0.08 ^a^	0.13 ± 0.15 ^b^
C22:1 (erucic acid) ω9	0.25 ± 0.05 ^b^	0.3 ± 0.04 ^a^	ND	ND
Total MUFA	51.38 ± 0.49 ^a^	49.53 ± 1.42 ^b^	45.27 ± 0.65 ^d^	47.65 ± 1.74 ^c^
**Polyunsaturated fatty acids**				
C18:2n6 (linoleic acid) ω6	34.48 ± 0.83 ^d^	36.57 ± 1.29 ^b^	37.3 ± 0.4 ^a^	35.23 ± 5.8 ^c^
C18:3 (linolenic acid) ω3	6.58 ± 0.78 ^a^	5.87 ± 1.33 ^a^	3.35 ± 0.12 ^b^	6.84 ± 0.44 ^a^
C20:2 (eicosadienoic acid)	ND	ND	0.04 ± 0.05	ND
Total PUFA	41.06 ± 1.61 ^a^	42.44 ± 2.62 ^a^	40.69 ± 0.57 ^a^	42.07 ± 6.24 ^a^
Total UFA	92.44 ± 2.1 ^a^	91.97 ± 4.04 ^a^	85.96 ± 1.35 ^a^	89.72 ± 7.98 ^a^
**Relations**				
SFA/UFA	0.081	0.086	0.154	0.114
Essential fatty acids (ω6 + ω3)	41.06	42.44	40.65	42.07
Cox value	5.42	5.46	4.96	5.50
PUFA/ SFA	5.43	5.35	2.90	4.09
PUFA/ MUFA	0.79	0.85	0.89	0.88
UFA/SFA	12.23	11.58	6.12	8.73
L/O	0.68	0.76	0.87	0.76
Ln/O	0.13	0.12	0.07	0.15
L/Ln (w6/w3)	5.53	6.23	11.16	6.8

FA = fatty acids, SFA = saturated fatty acids, MUFA = monounsaturated fatty acids, PUFA = polyunsaturated fatty acids, UFA = unsaturated fatty acids, O = oleic acid, L= linoleic acid, Ln = linolenic acid. The values with a different letter in the same row had significant differences. ND = not detected.

**Table 3 plants-11-02301-t003:** Bioactive compounds and antioxidant activity of unroasted and roasted sweet and bitter lupin seeds oil.

Lupin Seed Oils				
Bioactive Content	Sweet Lupin Seeds Oil		Bitter Lupin Seeds Oil	
	Unroasted	Roasted	Unroasted	Roasted
**Phytosterols (mg/100 g)**				
Campesterol	74.61 ± 1.57 ^b^	70.42 ± 1.43 ^c^	77.35 ± 3.54 ^a^	65.85 ± 2.1 ^d^
Stigmasterol	0.93 ± 0.07 ^b^	0.03 ± 0.05 ^d^	1.35 ± 0.09 ^a^	0.72 ± 0.15 ^c^
β-sitosterol	326.57 ±3.91 ^a^	306.76 ± 3.61 ^b^	303.8 ± 6.10 ^b^	270.62 ± 8.7 ^c^
Δ-5 avenasterol	7.91 ± 0.22 ^c^	5.34 ± 0.39 ^d^	59.13 ± 5.9 ^a^	45.95 ± 2.15 ^b^
Brassicasterol	0.39 ± 0.15 ^b^	0.1 ± 0.01 ^d^	0.96 ± 0.18 ^a^	0.27 ± 0.25 ^c^
Total	406.18	382.65	442.59	383.41
**Carotenoids (mg/kg)**				
Lutein	0.65 ± 0.15 ^a^	0.17 ± 0.05 ^c^	0.31 ± 0.03 ^b^	0.07 ± 0.05 ^d^
Zeaksantyna	1.24 ± 0.17 ^a^	0.77 ± 0.15 ^b^	0.03 ± 0.05 ^c^	ND
β-Carotene	99.3 ± 0.36 ^a^	97.97 ± 0.6 ^b^	66.57 ± 0.68 ^c^	65.57 ± 0.35 ^d^
**Antioxidant activity**				
AOA %	74.22	76.7	66.63	72.89
EC_50_ (mg oil)	43.74	30.17	62.68	41.34
1/EC_50_x1000	23	33	16	24

TPC = total phenolic content (µg Gallic acid/1 g oil), AOA% = antioxidant activity (value represents the percent inhibition of oxidation of the linoleic acid/β-carotene emulation), EC_50_ = concentration of extract that causes a 50% decrease in DPPH absorbance, 1/EC_50_ = antiradical power. The values with a different letter in the same row had significant differences. ND = not detected.

**Table 4 plants-11-02301-t004:** Screening antibacterial and antifungal properties of unroasted and roasted sweet and bitter lupin seeds oil against microbial pathogen strains.

	Sweet Lupin Seeds Oil		Bitter Lupin Seeds Oil		Reference Standard
	Unroasted	Roasted	Unroasted	Roasted	
**Antibacterial activity** **(IZ; mm)**					**Ofloxacin**
*Salmonella senftenberg* *ATCC 8400*	3 ± 0.57 ^e^	4.67 ± 0.33 ^d^	6.33 ± 0.67 ^b^	5.33 ± 0.33 ^c^	23.7 ± 0.12 ^a^
*Escherichia coli* *ATCC 11228*	4.0 ± 0.57 ^e^	4.33 ± 0.67 ^d^	5.67 ± 0.33 ^c^	6.33 ± 0.33 ^b^	20.4 ± 0.58 ^a^
*Staphylococcus aureus* *ATCC 33591*	7.33 ± 0.33 ^d^	6.33 ± 0.67 ^e^	9.67 ± 0.88 ^b^	9.33 ± 0.12 ^c^	22.67 ± 0.33 ^a^
*Bacillus subtilis DB 100*	6.67 ± 0.33 ^d^	5.67 ± 0.33	10.33 ± 0.67 ^b^	8.67 ± 0.33 ^c^	22 ± 0.57 ^a^
**Antifungal activity** **(IZ; mm)**					**Nystatin**
*Aspergillus flavus* ITEM 698	4.33 ± 0.33 ^d^	3.33 ± 0.33 ^e^	6.67 ± 0.88 ^b^	5.67 ± 0.67 ^c^	26.2 ± 0.14 ^a^
*Aspergillus parasiticus* ITEM 11	6.33 ± 0.67 ^b^	4.33 ± 0.67 ^d^	5.67 ± 0.33 ^c^	4.33 ± 1.45 ^d^	24.7 ± 0.88 ^a^
*Aspergillus carbonarius* *ITEM 5010*	7.67 ± 0.67 ^c^	3.33 ± 0.33 ^e^	8.67 ± 0.33 ^b^	5.33 ± 0.67 ^d^	26.8 ± 0.58 ^a^
*Aspergillus ochraceus* ITEM 282	6.33 ± 0.33 ^d^	5.33 ± 0.67 ^e^	8.33 ± 0.67 ^b^	7.00 ± 0.58 ^c^	27.4 ± 0.67 ^a^
*Fusarium culmorum* KF 846	7.33 ± 0.33 ^d^	6.67 ± 0.33 ^e^	11.33 ± 0.58 ^b^	9.67 ± 1.15 ^c^	26.9 ± 0.74 ^a^

The results are expressed as mean ± SEM. Standard antibiotic (ofloxacin) was applied as a reference antibiotic for comparing the antibacterial effect. Standard fungicidal (Nystatin) was applied as a reference fungicidal for comparing the antifungal effect. The values with different letters in the same row had significant differences.

## Data Availability

All data regarding this work are represented inside this manuscript.

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
