# Peer review of "A Comprehensive Study of Lupin Seed Oils and the Roasting Effect on Their Chemical and Biological Activity"

_plants, 2022, doi:10.3390/plants11172301_

Round 1

Reviewer 1 Report

Your manuscript on lupin seed oils, their chemical and biological activity and the effects of roasting is timely and inspiring. I suggest the following revisions:

- Please add the results of statistical analysis to the tables and discussion. Express clearly which differences between the samples, or treatments, are statistically significant.

- Please add standard deviations to the results on Figures 1 and 2. Also, statistically significant differences should be expressed.

- On the row 16: "...(sweet and bitter)" refers now to biological activities. Please modify the sentence to express that sweet and bitter lupins were studied.  

Author Response

Point 1: Your manuscript on lupin seed oils, their chemical and biological activity, and the effects of roasting are timely and inspiring. I suggest the following revisions:

Response 1: Thanks, we will do that.

Point 2: - Please add the statistical analysis results to the tables and discussion. Express clearly which differences between the samples, or treatments, are statistically significant.

Response 2: thanks, these points were covered.

Point 3: - Please add standard deviations to the results in Figures 1 and 2. Also, statistically, significant differences should be expressed.

Response 3: thanks, these points were covered.

Point 4: - In row 16: "...(sweet and bitter)" refers to biological activities. Please modify the sentence to express that sweet and bitter lupin were studied.  

Response 4: thanks, these points were corrected.

Reviewer 2 Report

Overall, the figures and tables in this article must be rearranged. For example, What does "relation" in Table 2 mean? What are the units of the other values? What is the unit of the Y-axis in Figure 2? All data must be statistically analyzed, are there any significant differences? Activity assays require control compounds.

Author Response

Point 1: Overall, the figures and tables in this article must be rearranged. For example, What does "relation" in Table 2 mean? What are the units of the other values? What is the unit of the Y-axis in Figure 2? All data must be statistically analyzed, are there any significant differences? Activity assays require control compounds.

Response 1: These points were covered, thanks.

Reviewer 3 Report

The manuscript of Al-Amrousi et al. is an interesting experiment showing lupin seed oils roasting and unroasted and their effect on chemical and biological activity. My suggestions relate mainly to presentation of results.

Line 85-87. Please define what is roasted and unroasted. “… unroasted  sweet  lupin  oil  (USLO),  roasted  and  unroasted, …”. Confuse.

The acronyms are presenting confusion in identify the differences among extracts. Please, improve changes in paragraphs. I suggest the author’s emphasis in only one extract, in some cases, to avoid constant comparison.

For example Line 98-100:

Before: “The most abundant SFA are palmitic and stearic acids, which are higher in the UBLO (8.89 and 3.52%, respectively) than in the USLO (6.26 and 0.61 %, respectively).”

Change for: “The most abundant SFA are palmitic and stearic acids, with higher concentrations in UBLO (8.89 and 3.52%, respectively).”

Table 3. Campasterol have same values in roasted and unroasted sweet lupine oil.

In supplemental add the methodology and chromatograms of GC-Ms and HPLC in the respective wavelength.

Author Response

Point 1: The manuscript of Al-Amrousi et al. is an interesting experiment showing lupin seed oils roasting and unroasted and their effect on chemical and biological activity. My suggestions relate mainly to the presentation of results.

Response 1: Thanks for your time and the presented efforts

Line 85-87. Please define what is roasted and unroasted. “… unroasted  sweet  lupin  oil  (USLO),  roasted  and  unroasted, …”. Confuse.

The acronyms are presenting confusion in identify the differences among extracts. Please, improve changes in paragraphs. I suggest the author’s emphasis in only one extract, in some cases, to avoid constant comparison.

For example Line 98-100:

Before: “The most abundant SFA are palmitic and stearic acids, which are higher in the UBLO (8.89 and 3.52%, respectively) than in the USLO (6.26 and 0.61 %, respectively).”

Response 2: these points were covered, thanks

Change for: “The most abundant SFA are palmitic and stearic acids, with higher concentrations in UBLO (8.89 and 3.52%, respectively).”

Response 3: these points were covered, thanks

Table 3. Campasterol have same values in roasted and unroasted sweet lupine oil..

Response 4: this was a typographic mistake, and it was corrected, thanks.

Round 2

Reviewer 1 Report

Dear Authors,

Your manuscript is timely and interesting. After the major revisions you have done, the manuscript is now clear and comprehensive.  

Reviewer 2 Report

Figure 1 should not be connected by slick lines, but by straight lines.

Figure Lack of statistical analysis, Y-axis units